# Non-Parametric Evaluation Methods of the Brain Activity of a Bottlenose Dolphin during an Assisted Therapy

**DOI:** 10.3390/ani11020417

**Published:** 2021-02-06

**Authors:** Jesús Jaime Moreno Escobar, Oswaldo Morales Matamoros, Erika Yolanda Aguilar del Villar, Ricardo Tejeida Padilla, Ixchel Lina Reyes, Brenda Espinoza Zambrano, Brandon David Luna Gómez, Víctor Hugo Calderón Morfín

**Affiliations:** 1Escuela Superior de Ingeniería Mecánica y Eléctrica, Instituto Politécnico Nacional, 07340 Ciudad de México, Mexico; omoralesm@ipn.mx (O.M.M.); eaguilard2000@alumno.ipn.mx (E.Y.A.d.V.); ilinar1600@egresado.ipn.mx (I.L.R.); bespinozaz1400@alumno.ipn.mx (B.E.Z.); blunag1401@alumno.ipn.mx (B.D.L.G.); 2Escuela Superior de Turismo, Instituto Politécnico Nacional, 07630 Ciudad de México, Mexico; rtejeidap@ipn.mx; 3Delfiniti S.A. de C.V., 40880 Ixtapa Zihuatanejo, Mexico; vhcm1000@gmail.com

**Keywords:** biomedical signal acquisition, neurodynamical response, bottlenose dolphin, Dolphin-Assisted Therapy, TGAM1, BCI, EEG, FFT-PSD, Self-Affine

## Abstract

**Simple Summary:**

Dolphin Assisted Therapies (DAT) can be used with any person or group with specific needs and it can be as disparate as people at risk of social exclusion, eating disorders, terminally ill, mental health disorders, among many others. This paper is focused on measuring and analyzing dolphins brain activity when DAT is taking place, in order to identify if there is any differences in female dolphin’s neuronal signal when it is interacting with control or intervention subjects. In addition, we designed a wireless and portable electroencephalographic single-channel signal capture device to monitor the brain activity of a female bottle-nose dolphin. Our findings also validate the evidence that the interaction between a patient with a certain disease or disorder and undergoes to a DAT modifies usual brain activity behavior of a female bottle-nose dolphin.

**Abstract:**

Dolphin-Assisted Therapies (DAT) are alternative therapies aimed to reduce anxiety levels, stress relief and physical benefits. This paper is focused on measuring and analyzing dolphins brain activity when DAT is taking place in order to identify if there is any differences in female dolphin’s neuronal signal when it is interacting with control or intervention subjects, performing our research in Delfiniti, Ixtapa, Mexico facilities. We designed a wireless and portable electroencephalographic single-channel signal capture sensor to acquire and monitor the brain activity of a female bottle-nose dolphin. This EEG sensor was able to show that dolphin activity at rest is characterized by high spectral power at slow-frequencies bands. When the dolphin participated in DAT, a 23.53% increment in the 12–30 Hz frequency band was observed, but this only occurred for patients with some disease or disorder, given that 0.5–4 Hz band keeps it at 17.91% when there is a control patient. Regarding the fractal or Self-Affine Analysis, we found for all samples studied that at the beginning the dolphin’s brain activity behaved as a self-affine fractal described by a power-law until the fluctuations of voltage reached the crossovers, and after the crossovers these fluctuations left this scaling behavior. Hence, our findings validate the hypothesis that the participation in a DAT of a Patient with a certain disease or disorder modifies the usual behavior of a female bottle-nose dolphin.

## 1. Introduction

According to Maujean et al. in [1], Animal Assisted Therapies (AAT) have been used to improve the life-quality of patients with physical diseases and neurological disorders. To perform AAT, there have been used different domestic animals such as dogs, cats, dolphins, horses, birds, rodents, etc., yielding good outcomes in patients (children, adults or elderly people). Specifically, Cai et al. in [2] highlight that dolphins’ intelligence allows them to perform complex behavioral sequences and maintain interest in certain task when they participated in therapy. Additionally, they indicate that dolphin’s aquatic environment has been an excellent tool for improving in humans the motor skills and readiness to learn, provides kinesthetic feedback, increases cardiovascular and respiratory efficiency, restores cognitive and sensory motor perceptual patterns, greater flexibility in movement, decrease pain, and ease balance.

Liling et al. in [3] indicates that Bottle-nose dolphins are found in temperate and tropical waters around the world. They can inhabit different marine and coastal ecosystems of the Pacific, Atlantic and Indian Oceans. Their average length is between 2.5 and 2.7 m, the variation is due to the temperature of the water where they grow up, the colder the lengthier.

In the opinion of Reynolds et al. in [4], the maximum weight of the dolphins found in the Atlantic Ocean region was 284 kg. The bottle-nose dolphins’ auditory system includes a biological sonar ability, known as echolocation. They produce sounds at high frequencies that travel through the water, colliding with an object, and return to the dolphins in echo form. These waves are received through the cavities of the lower jaw, which are filled with fat to the middle ear, the inner ear and later to the auditory centers of the brain. By receiving echoes, these mammals can determine the size, shape, structure, composition, and direction of the object even if they are far away from 70. Moreover, Zhang et al. in [5] find that dolphins can vary the frequency of their sonar, depending on the area in which they are living. The frequencies of the sounds emitted by them range from 20–30 to 120–150 kHz.

According to Chengwei et al. in [6], dolphins appear to sense electrical fields from humans and attempt to communicate using the same frequencies, which generate several suppositions: (i) dolphin acoustic emissions, recorded in sea water, may bring about modifications in human brainwave activity, (ii) dolphin interaction may give patients pain relief due to the increased release of hormones (i.e., endorphin) into the blood, (iii) dolphin interaction may produce complex neurological, (iv) stimulation helping relaxation, reducing stress levels and, thereby, strengthening the immune system, and (v) dolphin’s ultrasonic energy may cause significant cellular changes within the living tissue of the central nervous system.

Birch, B. and D. in [7,8] suggest that DAT has been developed as an alternative treatment for people with diverse psychological and physical disabilities or disorders like Autism, Attention Deficit, Down Syndrome, Spastic Cerebral Palsy, and Obsessive Compulsive Disorder. As well as DAT aims to improve functioning through complementing and reinforcing existing therapies, rather than replacing them.

In agreement to Nathanson et al. in [9], a DAT has three theoretical bases: (i) an hypothesis that explains why people with different abilities find difficult to learn some daily activities, (ii) the main response to animal conditioning that explains the procedure, and (iii) the disciplinary team that will give the therapy to the patient.

Additionally, Stumpf and Breitenbach and Marino and Lilienfeld in [10,11] suggest that DAT has diversity of the experimental design, varied beliefs among what elements of the therapy work (i.e., dolphin, water, therapist), lack of regulation, the kind of approach to quantify several aspects that are going to be registered, types of diseases and disabilities (e.g., down syndrome, cerebral palsy, stress, language disabilities, speech, writing, autism, chronic pain, brain injuries, depression), characteristics of the population subject to study, the use of linear statistical tools for the analysis of the data collected, and usually its success is based on the perception of family members or people close to those who have been treated.

Research in DAT by Nathanson and de Faria, Nathanson et al., Johannes et al., Antonioli and Reveley, Iikura et al., N and Webb and Drummond in [9,12,13,14,15,16,17] seeks to demonstrate their effectiveness and efficiency on human conditions, but it does not include the effects on dolphins (cognitive and physiological level), or if there is a difference in their behavior when interacting in healthy people and patients.

Since being a therapy without standards fully established by pathology, there are findings by Richard et al., Kreiviniene et al. and Griffioen and Enders-Slegers in [18,19,20] that have variations in the procedures applied to patients, the number of therapists and/or specialists, time and periodicity of the sessions, as well as the selection of the dolphin and type of training to achieve therapeutic effects.

A few articles have focused on explore if there is any changes on dolphins due to DAT. For example, in [21] the authors use interaction programs with healthy humans of various age-groups by applying behavioral sampling methods in order to determine whether or not there are changes in the way dolphins work with humans and their congeners, providing knowledge of what might be the best conditions and setting standards for trainers to follow, people involved in the interaction and the type of training and care necessary for each of the dolphins.

Also in Trone et al. [22], the research focus on behavior of three dolphins (two males and one female) which participated in human interaction programs offered by a facility was study. Their findings suggest that the participation of these three-dolphins did not lead to any physical or mental harm, and even showed a behavior where the game tended to dominate itself, showing psychological health and a tendency to voluntarily participate in interactions with humans. Furthermore, Salgueiro in [23] mention that no aggressive or repulsion behavior was detected towards the individual and/or alterations in the welfare of dolphins. For that evaluation, protocols were used to quantify variations in motor, cognitive and developmental aspects of participants with autism spectrum disorders during the application of a program of interaction with dolphins.

Based on the above, research in DAT has been mainly focused on proof this kind of therapy works for treatment of several diseases in humans beings and dolphins ever suffered any harm or long term consequences due to their participation on DAT.

Brensing and Linke in [24] observed several untrained bottle-nose dolphins when these mammals interacted with assorted groups of swimmer, detecting an assisting behavior for children with mental and or physical disabilities. Their findings support our hypothesis that there is a change in dolphins when they participate in DAT. Hence, this work uses electroencephalography to collect information, and later process it by means of non-linear statistical tools, in order to insight if there is any change in the dolphins’ brain activity when interacting with healthy people and patients while DAT is taking place.

This paper is divided as follows. In Section 2 is shown basic concepts of electroencephalography to design a wireless and portable EEG signal capture device for a single channel, in order to record the dolphin brain activity. In Section 3 samples of a female dolphin-nose At rest are captured and Power Spectral Density and Self-Affine tools are proved to measure the previously collected samples. In Section 4 is presented the measurement of the dolphin’s brain activity when this cetacean participated in a DAT. In the next section, a General Discussion about our findings was made. Finally, in the last section was presented the Conclusions and Future work.

## 2. Materials and Methods

The design of a device to measure the dolphin’s brain activity requires a methodology where electroencephalography is first defined and studied to understand the signals to be treated, then the electroencephalographic device is designed and later is tested in a female bottle-nose dolphin. Electroencephalography is a tool designed to measure the electrical potentials produced by brain activity. That measurement is made with the help of electrodes placed on the scalp of the person to whom the Electroencephalogram (EEG) will be performed [25]. The EEG today is very useful for medical diagnoses since it helps the monitoring of some pathologies such as epilepsy, changes in the state of consciousness, infections of the central nervous system, etc. The amplitudes of the EEG signals range from 30 to 100 μV. There are different types of electrodes varying according their assembly. The basic assemblies are bipolar (transverse and longitudinal) and monopolar (referential). The bipolar mount records the voltage difference existing between two electrodes placed in areas of brain activity, while monopolar mount records the difference in potential between one electrode placed in an active brain area and the other placed in an area without activity or neutral [26]. Each electrode measures the electrical potential with respect to the reference electrode. The EEG can be performed using different types of electrodes, being surface electrodes, semi-invasive electrodes or intracranial electrodes [27].

In particular, Cencen et al. in [27] mention that EEG is carried out with surface electrodes and using two different methodologies to carry it out: (i) Standard EEG, this is performed with the patient at rest, activation maneuvers are performed such as photo stimulation at different frequencies. Its realization lasts approximately 30 min. (ii) EEG with sleep deprivation, this is performed with the patient sleep deprived naturally, one night before patient must stay awake and many hours before the EEG, without any stimulant to help the patient to stay in that state. Hashio et al. in [28] studied bottle-nose dolphins (*tursiops truncatus*), finding that the daily state or common behavior of these cetaceans was based on a prevalence of EEG frequencies within 8–12 Hz band, and in certain periods of time also 4–8 Hz band and even 0.5–4 Hz band are presented, i.e., dolphins are kept in a state of relaxation and tranquility [29].

In this way, for the development of the Electroencephalographic device used to read the data coming from ThinkGear ASIC Module EEG Sensor (TGAM1, Figure 1a), adjustments must be made both in hardware and software, that is why two versions of EEG devices are presented. The development of the prototype can be divided into two main reconfiguration steps: (i) Physical reconfiguration describes the main measurement electrode, the cables are from the ground and reference electrodes, as well as the shields of the system electrodes to attenuate as much as possible the environmental noise. (ii) Logical reconfiguration is required for sensor manipulation and communication in the use of libraries and licenses provided by the manufacturer *NeuroSky*. These libraries contain the necessary code to extract the data received at the moment of physical contact between the device and midline sagittal area of dolphin brain or $P_{Z}$ electrode, explained forward in this section.

In addition, Physical Reconfiguration is made using a TGAM1 sensor for acquisition of dolphin brain signals and a Bluetooth HC06 B26782h module for wireless data transmission. This sensor has different connections, but among the most important are the power, transmission, reception, and electrode connections [30]. TGAM1 EEG Sensor uses a main electrode called EEG signal pad made of silver, which is a good conductor. The design of this sensor holds impedance below 5 KΩ in each electrode because both the sort the electrodes’ material and the dolphin’s skin under salt water can reduce the impedance and increase conductivity [30,31]. Ground electrodes and their function is to make a difference between EEG signal pad and the ambient noise. For store all the electronic components of the EEG device, a waterproof container is required, easy to handle and strong enough for safe and reliable handling. So a *Sonoff* type box was chosen with the following characteristics: Waterproof certification IP66, ABS, Transparent cover material, Fire Rating V0, with 132.2 × 68.7 × 50.1 mm and 145.0 g of weight.

In terms of data acquisition, TGAM1 sensor has a fixed sample rate of 512 samples per second, i.e., 512 Hz and 10 bit depth for EEG time series resolution. In addition, the communication rate of this module requires a baud rate of 9600 baud, and by default it is set to 57,600 bauds. On the chip, a regulator is used internally in order to meet the requirements of the CMOS technology and handle 3.3 V throughout the TGAM1 module, serving to power the Bluetooth module. Logical Reconfiguration is developed in order to verify that a device is connected to the communication port through Bluetooth protocol. If it exists, it executes the *CONNECTION* function, otherwise it executes *FAIL*, then it validates the device by means of the *VALIDATION* flag and verifies that Send the data being read as a result of the dolphin’s neuronal activity. Thus, by means of the *DATA RECEPTION* function, each data received serially is saved in a vector assigned to it and these are ordering data in a file with a *TXT* extension. After a few minutes of sample capture, communication port is closed and file is saved in the root of system hard drive [32].

Therefore, it was possible to design the device shown in Figure 1b. Active Electrode is placed on brain 6 cm from blowhole, since Supin et al. in [33] indicate that this distance has the highest voltage of the dolphin’s brain activity, while the reference is placed on the melon of this cetacean. At this stage of the project, the way in which the device would be handled was decided. To verify the connection quality, it is measured by the *PoorSignal* Flag, taking values from 0 to 200 and where 0 means an excellent connection and 200 means that the device’s EEG electrode came off the skin. Values less than 51 are accepted as valid and those over 51 should be discarded since they are not reliable samples because they contain a lot of noise. A Valid EEG measurement was not captured at the beginning of the sample, after a stabilization time the *PoorSignal* Flag took values less than 51, meaning both validity and strength of the EEG Signal. Distances between electrodes and references could be measured and with it the need to handle the device in a comfortable and easy way was found. For the first functional test of EEG device, i.e., a control test, a female bottlenose dolphin named *Kaly* was carried out, whose characteristics will be described later. The taking of the first samples was obtained after *Kaly* obtained a positive reinforcement of her behavior, it could also be noted that it was carried out at noon, this time is the most common for the performance of assisted therapies. It is important to highlight, EEG time series were obtained on midline sagittal plane of dolphin skull. EEG power is dependent on active electrode position along midline sagittal area. EEG power is maximal in a position ranging from 6 to 9 cm from blowhole or $D_{bh}$. In this area, EEG waves have their maximum power [33]. To ensure distance $D_{bh}$ and always measure along midline sagittal line, active, reference and ground electrodes were placed along a semicircle with a difference in distance, i.e., distance between center of box and active electrode is 17 cm, while from the container center to reference and ground electrodes it is 22 cm. Then, center of box where circuits are located is aligned both vertically and horizontally with center of blowhole, thus reducing measurement error of brain activity in another part of dolphin’s skull. This point will be defined as the position of the active electrode or $P_{Z}$. Figure 1c shows that the dolphin had to be trained to first accept the EEG device and another training session for the dolphin to be allowed to take the measurement during the DAT. Once the dolphin accepts this device, verification of the connection quality using the *PoorSignal* Flag was proved, where at the beginning of the sample a valid EEG measurement was not captured, after a stabilization time the *PoorSignal* Flag took values less than 51, meaning both validity and signal strength EEG.

Once the prototype is tested, we proceed to eliminate undesirable signals that could be included in the sample collection by means of digital filtering. This procedure is capable of attenuating certain spectrum frequencies of the input signal and allowing the others to pass by. These filters carry digital components. The analog EEG signal is converted to digital using an A/D conversion system such as TGAM1. Undesirable frequencies are attenuated in the rejection band. The desired frequencies are transmitted and are found within the bands on either side of the notch. Generally, the gain of the passband of the suppressor filters is 1 or 0 db. Thus, Notch Filter allows the passage of the lower or higher frequencies to two determined frequencies, called lower cut ($fc_{1}$) and higher ($fc_{2}$), respectively. Frequencies in the band delimiting $fc_{1}$ and $fc_{2}$ are attenuated. In applications where low-level signals need to be amplified, there is a possibility that there may be one or more undesirable noise signals. For example, the 50, 60 or 400 Hz frequencies of power supply lines; the 100 Hz ripple produced by full-wave rectifiers, or even higher frequencies generated by switched regulated power supplies or clock oscillators. If both signals and a noise component of the same frequency are passed by the suppressor filter, only the desired EEG signals will exit the filter. The noise frequency is suppressed with the notch. This filter was digitally implemented in Matlab R2020b by using the *iirnotch* function, although the TGAM1 EEG Sensor has an analog filter to eliminate unwanted signals.

## 3. Experimental Design

### 3.1. Initial Conditions

The dolphins studied were female bottle-nose dolphins from the *Delfiniti* dolphinarium located in Ixtapa Zihuatanejo. *Delfiniti Ixtapa* facilities have three interconnected trenches four-meters deep, several platforms around the trenches and seven dolphins with an averaged age of eight-years old. In the most recent years, the activities of *Delfiniti Ixtapa* have been in four basic axes: (i) *Commercial Activities*: Experiences with dolphins that include swimming, encountering with families or children and of course shows where dolphins perform conditioned activities to entertain a resident group of people; these activities are the main source of resources for dolphinarium. (ii) *Educational Activities*: Facilities are continuously visited by elementary schools, where marine life specialists from dolphinarium explain to children principles of biology and respect for animal species; these activities do not generate any economic benefits for company, but they do generate social benefits for elementary schools in coastal region of Guerrero State, Mexico. (iii) *Therapeutic Activities*: Within Dolphin-Assisted Therapies (DAT), a program of five sessions with a 30-minutes duration each session is carried out; these therapies are performed depending on mobility skills, cognition and needs of each patient; this activity is sponsored by various companies with social responsibility and scholarships for low-income patients are included. DAT is aimed at patients with special needs such as Autism Spectrum Disorder, Down Syndrome, Cerebral Palsy, Attention Deficit Disorder, Language Disorders, Learning Disorders, Anxiety Disorders and Mood Disorders in teenagers. (iv) *Research Activities*: Since 2018 *Delfiniti Ixtapa* has been investigating the reason why DAT can have a therapeutic effect for some patients; within this dolphinarium trainers and therapists are trained because of their important role in DAT; and these activities have been reported by means of some publications such as [34,35,36]. The research resources are limited to the collaboration between *Delfiniti Ixtapa* providing the facilities and the National Polytechnic Institute of Mexico, in research projects with budgets less than $ 3000 USD per year.

This study was carried out on the female dolphin named *Kaly*, 21 years old and 225 kg of weight. There are two main reasons for employing only one dolphin: (i) Most dolphins are assigned to tasks related to commercial activities, and (ii) DAT require a dolphin not only to be female but also to be lactating, increasing its maternal instinct. *Kaly* was the only one who was accompanied by her offspring during the DAT, limiting the study to obtaining samples from this dolphin.

For performing effectively a DAT, cooperation of both parties is encouraged through rewards known as conditioned behavior. If proposed exercises are done correctly, patient is rewarded through caresses, games and activities with dolphin, while the latter’s cooperation is rewarded through attention and food as prizes. At *Delfiniti Ixtapa*, they have developed their own work methodology during therapy. Thus, in dolphin interaction, patient is oriented to caress it with complete movements that stimulate orientation and movement in three-dimensions: length, height and depth.

Therapist instructs trainer where to direct dolphin’s nose to specific points on patient’s body until a light touch is made. Starting in upper part of chest where thymus is located, a gland that develops T lymphocytes, responsible for cellular immunity of human body. Sonar emission of dolphin can be perceived by therapist, either by waves in water or by a sensation in area of body, where it is in contact. Next points of contact with dolphin are frontal lobe located on patient’s forehead, and later on nape where medulla oblongata is located, communicating peripheral nervous system and spinal cord with brain. Then, therapist leads dolphin to middle of head, in order to locate it in pituitary gland, an important gland for the endocrine system in charge of producing hormones that carry instructions and information between groups of cells. Dolphin is oriented towards occipitals and parietals of patient in a vertical position, where it is usual for certain discomfort to occur when dolphin is approached to left hemisphere of patient’s brain. Each of these positions has a maximum duration of one minute, there are also exercises where dolphin’s head is totally submerged. That is why the DAT duration depends on venue in which it takes place, in *Delfiniti Ixtapa* there are sessions 15 to 30 min, depending on specificity of each patient. Present proposal tries to make measurements with interfering human-dolphin interaction, hence we have records less than 2 min, since most of dolphin positions are very close to melon area of dolphin, in opinion of therapist changing routines would affect the performance of DAT. It is important to mention that the conditions of the tank or the time of day are not varied by recommendation of the Delfiniti technical staff due to the dolphin’s mood can change along the day, because of the fact that any female bottlenose dolphin is always accompanied by her calf during therapy. On the other hand, sample-taking test conditions were directly dependent on disposition to capture these samples. In this work the same position of dolphin was repeated for the same position in patient’s therapy.

Moreover, for placing the device on the dolphin, a 7-step methodology was followed:Started the dolphin training to become familiar with the device: Being a foreign object to the dolphin, the dolphin did not allow the device to be placed on it. As the trainer gave the instruction to approach by means of a whistle, little by little the dolphin began to bring the device closer to it until the dolphin had physical contact with it, each attempt gave the dolphin a prize and little by little the dolphin gained confidence. This procedure took approximately 20 min.Activate Bluetooth on the personal computer: At this point the Bluetooth of the computer must be activated to pair with the EEG device.Power the EEG device: Once the trainer had the confidence of the dolphin, the device was turned on through the switch positioned on the plastic casing, to proceed to take the measurements.Place the device on the dolphin’s head: Once the dolphin was confident, the trainer placed the device on its head while maintaining contact between the device and the dolphin until the measurement was completed, Figure 1c.Run the capture Library: The program that allows obtaining the samples from the device must be run, it searches for the connection with the port where the Bluetooth device is paired, and if it is not found, it sends an ERROR message and the program stops.Take brain activity for 2 min: Dolphins’ brain activity was taken for at least 2 min to ensure that the number of valid samples was sufficient.Store the raw samples for later use: After the 2 min were up, the measurements were saved in the root of local disk in a folder *C:\temp_DelfinitiEEG\*.

All the methodology was performed at the edge of the pit, since DAT are on a platform near the shore and the device has to be manipulated by the trainer.

### 3.2. Results and Discussion

For the development of a non-parametric evaluation of the brain activity of a bottlenose dolphin, this analysis had to be divided into three parts: (i) initial considerations of the stability of brain activity by estimating the Poor Quality Flag, (ii) a *Welch Power Spectrum Density Analysis* and thereby segment into different frequency bands, and (iii) a *Self-Affine Analysis* to know the fractal behavior of brain activity and what degree of correlations the electroencephalographic signal has.

The EEG device captured a series of measurements of the dolphin’s brain activity to be used and processed. Three measurements were made with a 2-min duration each one but only one minute of effective connectivity. Figure 2a shows an example of an original or RAW EEG Time Series.

All these EEG Time Series are reported as fluctuations in microvolts against the elapsed time of the sample, i.e., μV$v_{s}$ seconds. This work uses the following conversion equation of discrete EEG Time Series to μV [37]:(1)$$Voltage=EEG\;Time\;Series\times \frac{\frac{1.8}{4096}}{2000}\;\left[\mu V\right].$$

In literature there are works that explore Sensory Physiology of Aquatic Mammals such Supin et al. in [33], where smaller values are reported than those reported in this work. This is because these studies mainly include Hearing in Cetaceans, Pinnipeds and Sirenians, and Vision and Somatic Sense in Aquatic Mammals, i.e., these studies are more related to hearing of cetaceans than to neurodynamic response of their brain, as proposed in the present proposal.

TGAM1 EEG sensor has a flags system not only to analyze poor quality of electroencephalographic signals but also their validity. This is done by measuring and flagging certain undesirable characteristics in a valid brain activity when they are present, such as Signal Flatness ($SG_{flg}=25$), Signal Excessiveness ($SE_{flg}=26$), Power Ratio ($PR_{flg}=27$), and Off-Head Detection ($OHD_{flg}=29$). When none of these features is present, value is equal to zero. Thus, Equation Equation 2 defines Poor Quality Flag ($PQ_{flg}$) as follows:(2)$$PQ_{flg}=SG_{flg}+SE_{flg}+PR_{flg}+OHD_{flg}$$

Therefore, if there is a large muscle contamination in EEG activity or from rubbing of EEG electrode, $PQ_{flg}$ shows a unique combination that indicates grade of noise due to any perturbation. Hence, we only took into account samples where $PQ_{flg}$ met following requirements: (i) Values were different to 200 because they could indicate bad connection, and then data from sensor would be noise; (ii) Values less or equal to 51 indicate recorded data in a consistent way; and (iii) Values equal to 0 indicate an optimal connection.

Also, Figure 2a shows a measurement or sample of dolphin’s brain activity observed at rest and without a patient. Sample depicted in this figure indicates that in most of the total measurement were received voltage levels different than 0, i.e., in the periods when the dolphin was in contact with the device, there was a minimal lost of voltage. In sample some examples, it is observed that there were more disconnection periods in the middle of the measurement. And in some cases, there was disconnection and the power of the voltages captured was not as high as the other measurements. In general, the all three samples presented noise at the beginning because the TGAM1 sensor takes a transient connection time to consider the valid samples. Later, the samples begin to stabilize, however, when the connection is lost, some peaks are generated due to the unstable connection.

Besides, Figure 2b shows the amount of additive noise that was removed from the samples in the initial transients. All this was carried out using a notch filter, removing the 60 Hz-noise from the 120 volts power line for the case of Mexico, introducing the original EEG signal. In some cases the additive noise presented can be associated with the initial transient of any signal.

Subsequently, for analyzing the results obtained, the two non-parametric methods used were the *Welch Power Spectrum Density* and the fractal or *Self-Affine Analysis*.

The *Welch Power Spectrum Density* is a non-parametric tool, that is computationally efficient since they use the Fast Fourier Transform and, therefore, are suitable for exhaustive studies such as the presented in this work, where the lenght of time series reaches several minutes and the data storage space is high [38]. However, these methods estimate Power Spectral Density directly from the finite sample long without a model, yielding the effect of leakage and then frequency components can be masked. The Welch Periodogram uses a definition of the Spectral Power Density for ergodic processes based on a temporal averaging, i.e., its statistical properties can be determined by a single sufficiently long realization of the processed time series. This Welch Periogram or Sample Spectrum is calculated directly from the samples of the segment of the electroencephalographic time series.

The *Self-Affine Analysis* is another non-parametric method used in this work to study the stochastic or self-affine time series generated by complex systems over many time-scales. Altmann and Kantz in [39] found that self-affine time series exhibit long-term correlations decaying as a power-law or scaling behavior. It is worth to detect if time series yielded by complex systems display long-term temporal correlations because this sort of correlations indicates that even data points that are separated by a very long time still can have some statistical correlation between each other over many time-scales for a better understanding about the dynamics of the system under analysis. Many complex system can display short or long term correlations over several time-scales described by power-laws, where the scaling exponent quantifies the *sort and grade* of correlations [40]. When this happens, it is possible to make statistical predictions about the probabilistic future states of the system and how it could change. The statistical or function correlation can be quantified using tools from the theory of fractals such as the Self-Affine Analysis. When the correlation function is long-term, there is no characteristic correlation time, i.e., time series exhibit the fractal property of the scale-invariance symmetry or self-similarity. Thus, long-term correlated time series are also termed fractal or self-affine time series due to their self-similar behavior and typical *roughness*. One way to detect long-term correlations is studying how the observable variable (in this work, microvolts) fluctuates over time due to the observable variable by itself is unpredictable, i.e., it displays a random walk behavior or no correlations. However, it can be possible to find long-term correlations if we study the time series of fluctuations. Therefore, we propose to apply self-affinity or fractal analysis to study the time series of fluctuations of the dolphin´s brain activity, in order to find long-term correlated.

Hence, on the one hand, the spectrum of a signal is called its decomposition on an amplitude scale with respect to the frequency, and it is done by means of the Fourier series and represented by graphs called *Periodogram* or *Power Spectrum Density*. While the Original Time Series is a signal analyzed in relation to time, the *Periodogram* or *Power Spectrum Density* does it by means of the Fourier series, analyzing its Welch Power Spectrum Density.

Figure 3 shows the Average Power Spectrum Density, while Figure 4 shows Power Spectrum Density or Periodogram 0.5 to 256 Hz. These results indicate that the dolphin was relaxed since low-frequencies predominated in the three samples taken, which is consistent with the work done by Hashio et al. in [28], where the dominant-frequency in the bottle-nose dolphin’s brain activity is α. In Figure 3, the band predominating in the sample time is highlighted by means of a green box, thus in the three samples the band between 0.5 to 12 Hz has the highest Average Power Spectrum Density. For this experiment, we considered low-frequencies 0.5–12 Hz and high-frequencies 12–60 Hz. Separating the bands of dolphin’s brain activity as bands of human’s brain activity was for the sole purpose of illustrating the power in these bands, and we do not mean to say that they are equivalent.

On the other hand, a self-affine fractal is a set remaining invariant under an anisotropic scale transformation. Many real complex systems (e.g., the dolphin’s brain) can be treated as self-affine fractals in a way of long time series recorded. In this work we define a time series as one-dimensional array of numbers ($x_{i}$), i=1,…,N, representing values of an observable *x* usually measured equidistant in time.

Many self-affine time series yielded by complex systems display fluctuations on a wide range of time scales following a scaling relation over many time-scales described by fractal scaling exponents. Fractal or *Self-Affine Analysis* (SSA) has been applied to many biological time series for understanding their dynamics at different time-scales [41,42].

Another way of estimating whether there was a change in the bottle-nose dolphin’s brain activity during DAT, i.e., to know how the voltage values recorded in the EEG change or fluctuate over time, was to perform SSA to the time series of the voltage fluctuations. Standard deviation has been the most used parameter to calculate fluctuation of a variable over time. Hence, from the original time series we generated the time series of voltage fluctuations for each experiment, including the Design Experiment.

After performing SSA, we obtain the value of the fractal scaling exponent, called the Hurst exponent (*H*), over many realizations of the time window of different sizes, in order to determine the type and grade of correlations displayed by the time series of fluctuations, F(t+τ), from the bottle-female dolphin’s brain activity when it was in contact with different patients underwent a DAT.

To carry out the SSA to calculate the scaling exponent *H*, we propose to estimate the structure function, $\sigma \left(\tau ,\delta_{t}\right)$, of the time series F(t+τ) due to structure function allow to determine if there is long-term memory (or correlations) in the time series F(t+τ) [43]
(3)$$\sigma \left(\tau ,\delta_{t}\right)=\bar{{\langle {\left[F\left(t+\delta_{t},\tau \right)-F\left(t,\tau \right)\right]}^{2}\rangle }^{1/2}}$$
where the overbar denotes average over all *t* in time series of length T−τ (*T* is the length of original EEG time series x(t)) and the brackets denote average over different realizations of the time window of size δ*t*.

Moreover, Balankin in [43] claimed that the structure function of the time series F(t+τ) could exhibit the following power law-behavior
(4)$$\sigma \;\propto \;{\left(\delta_{t}\right)}^{H}$$
where *H* is the Hurst exponent, which points the sort and grade of self-affinity or correlations are displayed by the time series F(t+τ).

The values of *H* are limited to 0<H<1. When H<0.5, the time series F(t+τ) display *antipersistent* behavior or negative correlations among their data, i.e., if the value of the observable variable is rising, it is more probable the next value is shorter, and vice versa. When H>0.5, they exhibit *persistent* behavior or positive correlations among their data, i.e., if the value of the observable variable is rising, it is more probable the next value is bigger, and vice versa. And when H=0.5, they show *random walk* behavior or zero correlations among their data, i.e., it is impossible to know the next value of the observable variable. In the last case is impossible to make predictions about the future sates of the system.

For this Experiment Design, we recorded time series of bottle-nose female dolphin’s brain activity, or original time series x(t), for three different measures (Figure 5a): the first measure (in red color) with length $T_{1}=10505$ voltage records (μVolts) versus time (seconds) for 20.52 s, the second one (in blue color) with length $T_{2}=10868$ (μV vs. sec) for 21.13 s, and the third one (in green color) with $T_{3}=10410$ (μV vs. sec) for 20.33 s.

Moreover, we constructed, from each time series x(t), 198 time series of voltage fluctuations, F(t+τ), i.e., we considered a range of the time interval of the sample of 3≤τ≤200, with time windows of the intervals samples $\delta_{t}=1/512$ seconds. For instance, see Figure 5b (for the first measurement), Figure 5c (for the second measurement) and Figure 5d (for the third measurement), all of them with τ=50.

In addition, a *crossover* is the point in time where the system whole behavior of the system changes; in our case, it was the τ where the time series F(t+τ) leave to follow a fractal behavior explained by a power-law: $\tau_{1}=192$ (for the first measurement), $\tau_{2}=117$ (for the second measurement) and $\tau_{3}=104$ (for the third measurement), see Figure 5 and Table 1.

From the time series F(t+τ) we performed the SSA, estimating the structure function $\sigma \left(\tau ,\delta_{t}\right)$ versus $\delta_{t}$ for 3≤τ≤200, for the three measures, in order to obtain the scaling exponent *H*, Figure 5e.

For the first measure (in red color), the value of the exponent *H* was $H_{1}=0.4775$, displaying negative correlations in the short-term or antipersistent behavior. For the second measure (in blue color), $H_{2}=0.2767$, displaying also antipersistent behavior. For the third measure (in green color), $H_{3}=0.4008$, displaying also antipersistent behavior, Figure 5e and Table 1.

Table 1 summarizes the most important results from Experimental Design, considered as the way to verify the dolphin’s brain activity before participating in a DAT. The slow-frequency band is greater than the fast-frequency band in the three samples. The range of Spectral Power Density for the frequencies ranges from $3.3\times 10^{7}$ to $5.1\times 10^{7}$dBHz, while frequencies range from $2.7\times 10^{7}$ to $3.9\times 10^{7}$dBHz. So the average for high frequencies is $3.43\times 10^{7}$dBHz, while for low-frequencies it rises 17.34%, reaching $4.06\times 10^{7}$dBHz. Hence, it can be asserted that the dolphin is in its relaxation state, typical of its condition in captivity.

Moreover, for the three measures, the time series F(t+τ) displayed an antipersistent behavior exhibited and characterized by the following power-law $\sigma \;\propto \;{\left(\delta_{t}\right)}^{H}$, Table 1, i.e., if the dolphin’s brain activity is rising, at rest and only in contact with its trainer, it is more probable the next voltage value was shorter, and vice versa. Perhaps, this antipersistent behavior was exhibited due to the adaptation of the dolphin to the designed device for this investigation.

## 4. Experimentation

### 4.1. Experimental Setup

The Experimetal Setup consisted in testing our EEG by means of capturing EEG signals from intervention patients with: (i) Pilocytic astrocytoma (Intervention Patient 1), (ii) Kinsbourne syndrome (Intervention Patient 2), and (iii) Infantile spastic cerebral palsy with epilepsy (Intervention Patient 3), as well as a Control Patient, when all of them underwent to a Dolphin-Assisted Therapy (DAT). Then, the whole system was subdivided into three fundamental parts:Female dolphin of bottle-nose species.Control or Intervention Patients.Dolphin EEG device v2.0.

The fundamental hypothesis is that the dolphin’s brain activity is affected, depending on whether it is a Control or Intervention Patient. This must happen because the first two subsystems interact with each other, while the third one registers that interaction to be analyzed. For studying dolphin’s brain activity, we decomposed the EEG signals into five fundamental Bands: 0.5–4 Hz, 4–8 Hz, 8–12 Hz, 12–30 Hz and 30–60 Hz by applying the fast Fourier transform (FFT), in order to estimate the Power Spectral Density, expressed in μVolts/Hz [44]. Then, we studied the dynamics of dolphin´s brain activity by using FFT and the Self-Affine Analysis from EEG signals [45]. The EEG RAW data were time series x(t) showing the dolphin´s brain activity (voltage vs. time) during a DAT (Figure 6) for all four patients, as recorded in the midline sagittal electrode ($P_{Z}$) by means of the Dolphin EEG device v2.0 Module.

The methodology used in this project was approved by the Ethics Committee of the National Polytechnic Institute of Mexico through Confidentiality Commitment Letter D/1477/2020. This document supports the way to collecting the several samples, as well as the treatment given to the bottle-nose dolphins (*Tursiops truncatus*) by the research team. Also, this project made a responsible use of the data from the patients, who were informed and gave their consent for the use of the data obtained in the experiments. Before taking the samples and connecting the devices, the patients were informed of the entire procedure; if anyone did not agree, they could immediately terminate their participation. PARTICIPANTS who agreed with the methodology and materials to be used to collect the samples signed a written informed consent form (25 January 2020). Also, PARTICIPANTS are informed of the tests that will be carried out on them, although some of them do not fully understand it, before entering the tank with the cetaceans. The time of year chosen was winter so that the temperature at noon during the test does not exceed 30 ${}^{\circ }$C, although the measuring equipment are kept at temperatures of no more than 20 ${}^{\circ }$C. After taking the sample, it is reported if the amount of information can be considered sufficient, if it is not enough, they are asked to stay a few more minutes in the tank to capture one more sample, no longer than 2 min.

We performed four experiments, carrying out DAT to the four following 12-year-old patients: (i) intervention patient with Pilocytic Astrocytoma, or Intervention Patient 1 (Experiment 1), (ii) intervention patient with Kinsbourne Syndrome, or Intervention Patient 2 (Experiment 2), (iii) intervention patient with Infantile Cerebral Palsy, or Intervention Patient 3 (Experiment 3), and (iv) control patient without any neurological disorder, or Control Patient (Experiment 4). For each experiment we made the following.

At the beginning, samples displayed only noise because of the time taken to establish contact between the electrodes and the dolphin. Then, a period of brain activity began and the measurement stars to be consistent after applying the notch filter, showing the voltage levels with respect to time. There were small periods where the voltages were almost zero continuously due to a period where the connection was lost, caused by the movement of the dolphin or the trainer, so the electrodes could stop having contact with the dolphin. See Figure 7a, Figure 8a, Figure 9a and Figure 10a.

Later, we elaborated a Periodogram representing the dolphin´s brainwaves during DAT for each patient. Depending on the patient, one or two samples were obtained with valid and consistent data in the same DAT. We focused on the 0.5–4 Hz and 12–30 Hz bands, thus the results from the power density were averaged in these bands.

We only took into account the samples recorded between the frequencies ranging from 0.5 to 60 Hz for disconnection periods, so the losses were minimal, almost imperceptible. In general, the signal was considered valid when $PQ_{flg}<51$, ensuring that signal power is the maximum and electrodes are at indicated places, although the signal is not entirely flat or excessive. The Average-Periodogram diagrams indicates the power at certain frequencies at a specific time, and only the periods when the connection was stable were considered.

Moreover, we performed the Self-Affinity Analysis (SSA) for the outcomes yielded in each experiment, from the time series of bottle-nose female dolphin´s brain activity x(t), collected when the dolphin was in contact with patients underwent a DAT. Each sample had different length of data (μVolts vs. seconds), depending on the time considered to realize the DAT. From each x(t), we constructed 198 time series of voltage fluctuations, F(t+τ), i.e., we considered a range of the time interval of the sample of 3≤τ≤200, with time windows of the intervals samples $\delta_{t}=1/512$ s. Finally, from time series F(t+τ), we estimated the structure function $\sigma \left(\tau ,\delta_{t}\right)$ versus $\delta_{t}$ for 3≤τ≤200, for the samples.

### 4.2. Experiment 1

In this experiment, a 12-year-old intervention patient with Pilocytic Astrocytoma, called the Intervention Patient 1, underwent a DAT. Pilocytic Astrocytoma is the most common primary brain tumor in children aged 0 to 19 years that can arise anywhere in the central nervous system, the most common symptoms depend on where it grows. When the cerebellum patients have ataxia, cranial nerve defects and signs of increased intracranial pressure (headache, nausea and vomiting); and when the tumor is located in the optical pathways, it produces loss of visual acuity or field defects; and when this tumor is located in the hypothalamus, it results in endocrine syndromes, such as diabetes insipidus, early puberty, or electrolyte imbalance [46].

Figure 7b shows an average Periodogram representing the dolphin´s brainwaves when Intervention Patient 1 underwent a DAT, for two samples during the same DAT. For 0.5–4 Hz band, it had a power of $0.65\times 10^{7}$ and $1.6\times 10^{7}$
dBHz, while 12–30 Hz band had $0.75\times 10^{7}$ and $2.0\times 10^{7}$
dBHz. Results from power density were averaged in 0.5–4 Hz and 12–30 Hz bands, yielding $1.13\times 10^{7}$ and $1.375\times 10^{7}$
dBHz, respectively, i.e., 12–30 Hz band was 22.22% higher. Hence, 12–30 Hz band was pointed out with a green rectangle, indicating that dolphin was concentrated with prevailing fast-frequencies. The interaction with this Intervention Patient 1 modified dolphin´s electroencephalographic behavior with respect to its original state, where slow-frequencies predominated. In Figure 7c Periodogram of activity frequency vs. power measured in dBHz is shown.

To perform the Self-Affinity Analysis (SSA), we yielded time series of bottle-nose female dolphin´s brain activity when it was in contact with the Intervention Patient 1 underwent DAT, x(t), for the two different samples in the same DAT (Figure 7a): Sample 1 (in red color) with $T_{S1}=18034$ data (μVolts vs. sec) for 35.22 s, and Sample 2 (in blue color) with $T_{S2}=4995$ data (μV vs. sec) for 9.76 s.

Subsequently, we constructed 198 time series of voltage fluctuations, F(t+τ), considering 3≤τ≤200 with $\delta_{t}=1/512$ s. For instance, see Figure 7d for Sample 1, with τ=50.

As we mentioned before, the crossover is the instant of time when the system dynamics leaves to behave as a self-affine fractal described by a power-law. The crossover for Sample 1 was $\tau_{S1}=114$ and for Sample 2 was $\tau_{S2}=157$, Figure 7e, pointing out changes in the dolphin’s brainwaves during the DAT.

Moreover, we estimated the structure function $\sigma \left(\tau ,\delta_{t}\right)$ vs. $\delta_{t}$ for 3≤τ≤200, for the two samples, (Figure 7e). For Sample 1 (in red color), $H_{S1}=0.5356$, it means that the 198 time series F(t+τ) display persistent behavior, i.e., if the fluctuations of the dolphin’s brain activity is rising, it is more probable the next voltage value is bigger, and vice versa. For Sample 2 (in blue color), $H_{S2}=0.5019$, it means that the 198 time series F(t+τ) display random walk behavior, i.e., the fluctuations of dolphin’s brain activity are unpredictable. As we can see, the value of *H* was reduced 6.29 % from Sample 1 to Sample 2 during the same DAT because Sample 1 lasted 25.46 more seconds respect to Sample 2.

### 4.3. Experiment 2

In this experiment, a 12-year-old intervention patient with Kinsbourne Syndrome, called the Intervention Patient 2, underwent a DAT. Kinsbourne syndrome, known for cerebellar opsoclonia-myoclonia-ataxia syndrome, is a rare neurological disease that mainly affects children (6 to 36 months) who were previously healthy. This disease is characterized by the presence of rapid, irregular, horizontal and vertical eye movements, myoclonias that can affect the trunk, limbs or face and ataxia cerebellar. Most patients suffer from persistent deficits in cognition, neurology, and behavior [47,48].

Figure 8b shows an average Periodogram representing dolphin´s brainwaves during DAT of Intervention Patient 2 for one sample. For 0.5–4 Hz band, it had a power of $1,1\times 10^{7}$dBHz, while 12–30 Hz band had $1.3\times 10^{7}$dBHz, i.e., 12–30 Hz band was 18.18% higher. That is why 12–30 Hz band was pointed out with a green rectangle, indicating that dolphin was concentrated, with prevailing fast-frequencies. Also, interaction with this Intervention Patient 2 modified electroencephalographic behavior of dolphin with respect to its original state, where slow-frequencies predominated. In Figure 8c Periodogram of activity frequency vs. power measured in dBHz is shown.

To perform the SSA, we yielded time series of bottle-nose female dolphin´s brain activity when it was in contact with the Intervention Patient 2 underwent a DAT, x(t), for a sample (Figure 8a) with $T_{=}5839$ data (μVolts vs. sec) for 11.40 s.

Later, we constructed from the time series x(t), 198 time series F(t+τ) with $\delta_{t}=1/512$ s. For instance, see Figure 8d, with τ=50. The crossover for the Sample was τ=144, pointing out changes in the dolphin’s brain activity, Figure 8e.

After estimating the structure function $\sigma \left(\tau ,\delta_{t}\right)$ versus $\delta_{t}$ for the Sample, Figure 8e, we obtained H=0.4880, it means that the 198 time series F(t+τ) display antipersistent behavior, i.e., if the fluctuations of the dolphin’s brain activity is rising, it is more probable the next voltage value is shorter, and vice versa.

### 4.4. Experiment 3

In this experiment, a 12-year-old intervention patient with Infantile Cerebral Palsy with epilepsy, called the Intervention Patient 3, underwent a DAT. Infantile Spastic Cerebral Palsy is a kind of Cerebral palsy (CP). CP is a non-hereditary neurological disorder that affect parts of the brain which control movement muscles in the body, and it appears in infancy or early childhood before 3-years old. CP spastic is the most common disorder among patients and it exhibits a wide variety of symptoms, including, lack of muscle coordination when performing voluntary movements, stiff or tight muscles and exaggerated reflexes, weakness in one or more arm or leg, walking on the toes, a crouched gait, or a scissored gait, variations in muscle tone, excessive drooling, difficulties swallowing or speaking, random involuntary movements, delays in reaching motor skill milestones and difficulty with precise movements [49]. In the case of epilepsy, it is a common chronic neurological disorder in which patients have seizures either involving both cerebral hemispheres or a focal originating from one cerebral hemisphere, resulting from abnormal, synchronized neuronal firing in the brain and encompasses a heterogeneous group of disease entities with diverse etiologies and outcome [50,51].

Figure 9b shows an average Periodogram representing dolphin´s brainwaves during DAT of Intervention Patient 3 for two samples. For 0.5–4 Hz band, it had a power of $2.5\times 10^{7}$ and $1.8\times 10^{7}$dBHz, while 12–30 Hz band had $3.1\times 10^{7}$ and $2.3\times 10^{7}$dBHz. Results from power density were averaged in 0.5–4 Hz and 12–30 Hz bands, yielding $2.15\times 10^{7}$ and $2.7\times 10^{7}$dBHz, respectively, i.e., 12–30 Hz band was 25.58% higher. Also, 12–30 Hz band was pointed out with a green rectangle, indicating that dolphin was concentrated, with prevailing fast-frequencies. The interaction with this Intervention Patient 3 also modified electroencephalographic behavior of dolphin with respect to its original state, where slow-frequencies predominated. In Figure 9c Periodogram of activity frequency vs. power measured in dBHz is shown.

To perform the SSA, we yielded time series of bottle-nose female dolphin´s brain activity when it was in contact with the Intervention Patient 3 underwent a DAT, x(t), for the two different samples, (Figure 9a): Sample 1 (in red color) with $T_{S1}=12527$ data (μVolts vs. s) for 24.47 s, and Sample 2 (in blue color) with $T_{S2}=25478$ data (μV vs. s) for 49.76 s.

Later, we constructed, from each time series x(t), 198 time series F(t+τ), with $\delta_{t}=1/512$ s. For instance, see Figure 9d for Sample 1, with τ=50. The crossover for Sample 1 was $\tau_{S1}=152$ and for Sample 2 $\tau_{S2}=91$, Figure 9e, pointing out changes in the dolphin’s brainwaves.

After estimating the structure function $\sigma \left(\tau ,\delta_{t}\right)$ versus $\delta_{t}$ for the two samples, Figure 9e, we obtained $H_{S1}=0.4537$ for Sample 1 (in red color) and $H_{S2}=0.4595$ for Sample 2 (in blue color). In both samples the 198 time series F(t+τ) displayed antipersistent behavior, and the value of *H* was raised 1.28 % from Sample 1 to Sample 2, because Sample 2 lasted 25.29 more seconds respect to Sample 1.

### 4.5. Experiment 4

In this experiment, a 12-year-old Control Patient without any neurological disorder, called the Control Patient, underwent a DAT.

Figure 10b shows an average Periodogram representing dolphin´s brainwaves during DAT of Control Patient for two samples. For 0.5–4 Hz band, it had a power of $1.75\times 10^{7}$ and $2.2\times 10^{7}$dBHz, while 12–30 Hz band had $1.55\times 10^{7}$ and $1.8\times 10^{7}$dBHz. Results from power density were averaged in 0.5–4 Hz and 12–30 Hz bands, yielding $1.98\times 10^{7}$ and $1.68\times 10^{7}$dBHz, respectively, i.e., in this case, 0.5–4 Hz band was 17.91% higher. Hence, 0.5–4 Hz band was pointed out with a green rectangle, indicating that dolphin was relaxed, with prevailing slow-frequencies. The interaction with this Control Patient did not modify electroencephalographic behavior of dolphin with respect to its original state. In Figure 10c Periodogram of activity frequency vs. power measured in dBHz is shown.

To perform the SSA, we yielded time series of bottle-nose female dolphin´s brain activity when it was in contact with the Control Patient underwent a DAT, x(t), for two different samples (Figure 10a): Sample 1 (in red color) with $T_{S1}=12733$ data (μVolts vs. sec) for 24,87 s, and the Sample 2 (in blue color) with $T_{S2}=12082$ data (μV vs. s) for 23.60 s.

Later, we constructed, from each time series x(t), 198 time series F(t+τ), with $\delta_{t}=1/512$ s. For instance, see Figure 10d for Sample 1, with τ=50. The crossover for Sample 1 was $\tau_{S1}=276$ and for Sample 2 $\tau_{S2}=183$, Figure 10e, indicating changes in the dolphin’s brainwaves.

After estimating the structure function $\sigma \left(\tau ,\delta_{t}\right)$ versus $\delta_{t}$ for the two samples, Figure 10e, we obtain $H_{S1}=0.5251$ for Sample 1 (in red color), it means that the 198 time series F(t+τ) display persistent behavior. For Sample 2 (in blue color), $H_{S2}=0.4890$, displaying antipersistent behavior. As we can see, the value of *H* was reduced 6.87 % from Sample 1 to Sample 2 because Sample 1 lasted 1.27 more seconds respect to Sample 2.

## 5. General Discussion

Table 2 summarizes the most important results yielded by the above four experiments carried out. The results of the Intervention and Control patients were ordered according to the Crossovers estimated from the Self-Affine Analysis.

Results of our Experimentation indicate a way to verify the dynamics of dolphins’ brain activity when they participate in a DAT. Regarding Power Density Analysis, on the average dolphin studied changed its common behavior from its condition in captivity by 10.30%. Now if samples from Intervention patients are averaged, we obtained that dolphin significantly changed its common behavior from its condition in captivity by 23.53%. When samples of Control Patient were separated, on average dolphin did not significantly changed its typical behavior from its condition in captivity, i.e., it kept its condition at 17.91%. These results clearly indicate that the participation in a DAT of a Patient with a certain disease or disorder modifies the usual behavior of a female bottle-nose dolphin, like the animal knew their condition.

After performing the Self-Affine Analysis to the time series of voltage fluctuations F(t+τ) for estimating their structure function, we found different behaviors during DAT: in Experiment 1 there was a transition from a persistent behavior to a random walk behavior, in Experiment 2 the behavior was antipersitent, in Experiment 3 the antipersistent behavior was hold, and in Experiment 4 there was a transition from antipersistent behavior to persistent behavior. These findings point out changes in the dolphins’ brain activity when they participate in a DAT, and these changes depend on the sort of patient and how last a DAT. However, in all these time series F(t+τ), at the beginning the dolphin’s brain activity behaved as a self-affine fractal described by power-laws over several time-scales until the fluctuations reached the crossovers; after the crossovers, the fluctuations left this scaling behavior. Besides, the sort of correlations displayed by the time series of fluctuations F(t+τ) were different, depending on how many time lasted each DAT.


Based on our findings, we consider that the Power Density Behavior of brain activity is modified, due to disturbances generated by the environment such as environmental conditions, as well as the physical and mental state of the participants involved during DAT.


## 6. Conclusions

This work presents the design of a wireless and portable electroencephalographic single-channel signal capture sensor to acquire and monitor the brain activity of a female bottle-nose dolphin. We did not aim to diagnose any disorder but only to record and measure the dolphin´s brainwaves during DAT by applying non-parametric methods.

Dolphins are not domesticated animals. Thus, the adaptation of these animals to the device developed required a period of learning and adaptation of dolphin, yielding some measurements to be erroneous due to the dolphin’s lack of physical contact. Measurements allowed the data collected to be processed not only by non-parametric methods such as the Power Spectrum Analysis and the Self-Affine Analysis but also by computational tools such as MatLab.

The experiments realized were compared by means of a *Welch Power Spectrum Analysis*, showing that the average power is greatly modified in the band of 0.5–4 Hz, when dolphin is in the presence of a neurotypical patient. In patients with neurological disorders underwent DAT, dolphin keeps a high-brain activity power in the 12–30 Hz band. Moreover, the slow-frequency band is greater than the fast-frequency band in the three samples studied when the dolphin showed its common behavior from its condition in captivity. On the average for high-frequencies is $3.43\times 10^{7}$dBHz, while for low-frequencies it rises 17.34% reaching $4.06\times 10^{7}$dBHz, suggesting that the dolphin is in its relaxation state, typical of its condition in captivity. When samples from Intervention patients are compared, on average, the dolphin significantly changed its common behavior from its condition in captivity by 23.53%, while samples from Control Patient samples are separated, on average, the dolphin did not significantly change its common behavior from its condition in captivity by 17.91%.

Regarding to *Self-Affine Analysis*, we have the following findings. For the Experiment 1, $H_{S1}=0.5356$ and $H_{S2}=0.5019$, with the crossovers $\tau_{S1}=114$ and $\tau_{S2}=157$, respectively; for the Experiment 2, H=0.4880, with τ=144; for the Experiment 3, $H_{S1}=0.4595$ and $H_{S2}=0.4537$, with $\tau_{S1}=91$ and $\tau_{S2}=152$, respectively, and for the Experiment 4, $H_{S1}=0.4890$ and $H_{S2}=0.5251$, with $\tau_{S1}=183$ and $\tau_{S2}=276$, respectively. These findings point out that at the beginning the dolphin’s brain activity behaved as a self-affine fractal described by the power-law $\sigma \;\propto \;{\left(\delta_{t}\right)}^{H}$ over several time-scales until the fluctuations reached the crossovers; after the crossovers, the fluctuations left this scaling behavior. Hence, these findings also validate the evidence that the participation in a DAT of a Patient with a certain disease or disorder modifies the usual behavior of a female bottle-nose dolphin.

According to the above findings, we validate our hypothesis the dolphin’s brain activity is modified when the cetacean is in contact with a patient with a mental or physical disease or disorder during DAT. Our work complements the research carried out by Brensing and Linke in [24], where the hypothesis of a change in dolphin behavior is also presented, but it is approached only from an pure observational point of view.

Besides, the sort of correlations displayed by the time series of fluctuations F(t+τ) were different, depending on how many time lasted each DAT. For example, in Experiment 1 the value of *H* was reduced 6.29% from Sample 1 to Sample 2 during the same DAT because Sample 1 lasted 25.46 more seconds respect to Sample 2; while in Experiment 3 *H* was raised 1.28% from Sample 1 to Sample 2, because Sample 2 lasted 25.29 more seconds respect to Sample 1. Based on these findings we could suggest that the more last the DAT, the more long-term correlations are displayed from the time series F(t+τ).

As future work to this proposal, activity of the dolphin brain can be correlated with the results detected with the electroencephalogram with other parameters related to stress using an electrocardiogram. Furthermore, the activity of the dolphin brain can be correlated with the brain activity of the patient via the study of the electroencephalogram in the patients. Finally, the present work can be validated using a well-described in literature paradigm such as a sensory evoked potential, or steady state response based on the works made by Supin et al. in [33].

## Figures and Tables

**Figure 1 animals-11-00417-f001:**
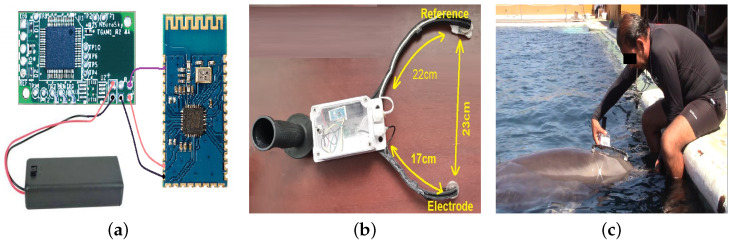
Design of EEG signal capture device adapted for a female bottlenose dolphin. (**a**) ThinkGear ASIC Module EEG Sensor (TGAM1). (**b**) EEG signal capture device. (**c**) Dolphin training for accepting the device.

**Figure 2 animals-11-00417-f002:**
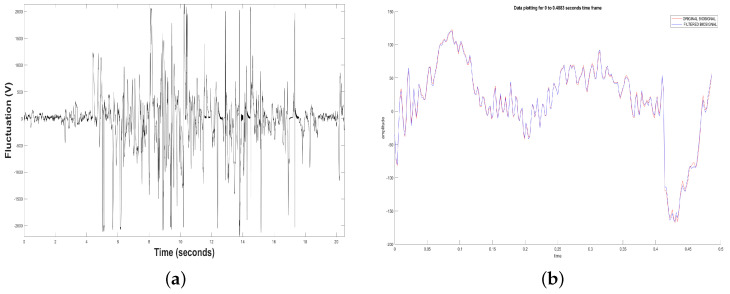
(**a**) Original Time Series. (**b**) Filtered Time Series by means of a notch filter.

**Figure 3 animals-11-00417-f003:**
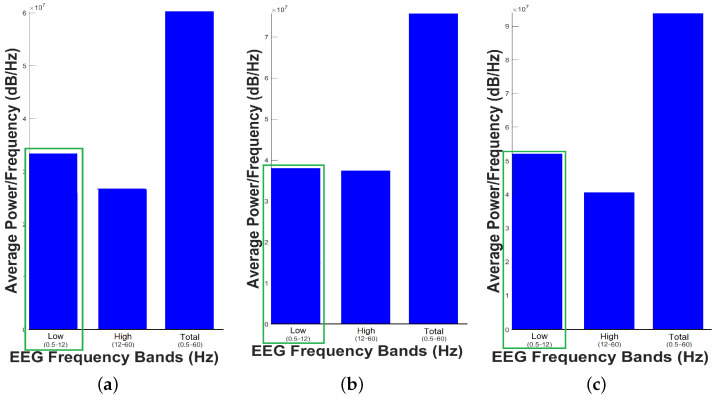
Average Power Spectrum Density. (**a**) First sample, (**b**) Second Sample and (**c**) Third Sample.

**Figure 4 animals-11-00417-f004:**
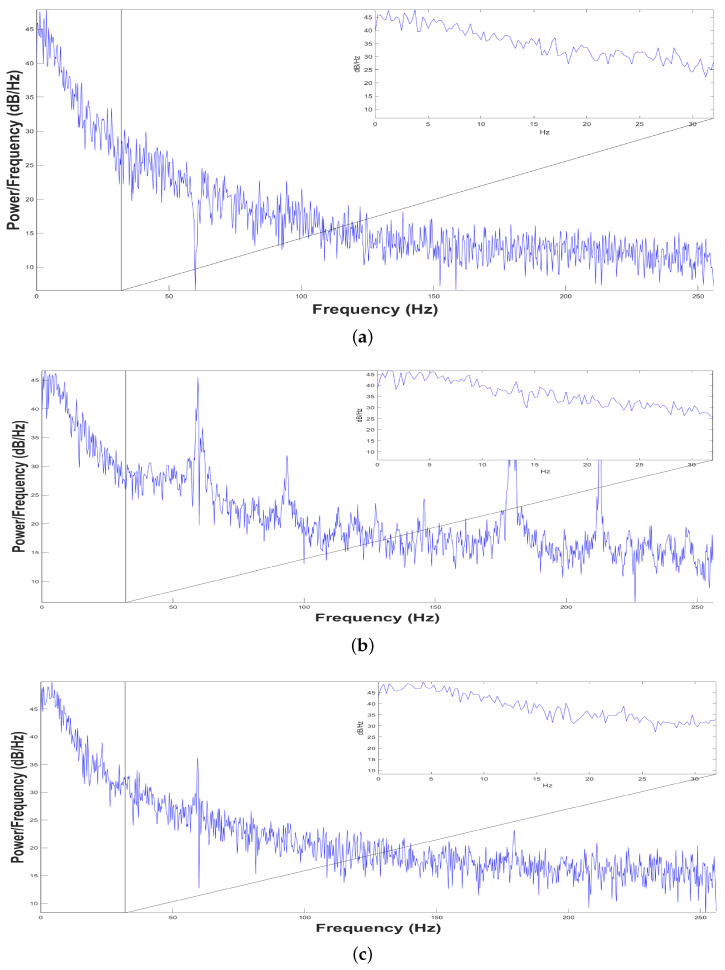
Power Spectrum Density from 0.5 to 256 Hz. (**a**) First sample, (**b**) Second Sample and (**c**) Third Sample

**Figure 5 animals-11-00417-f005:**
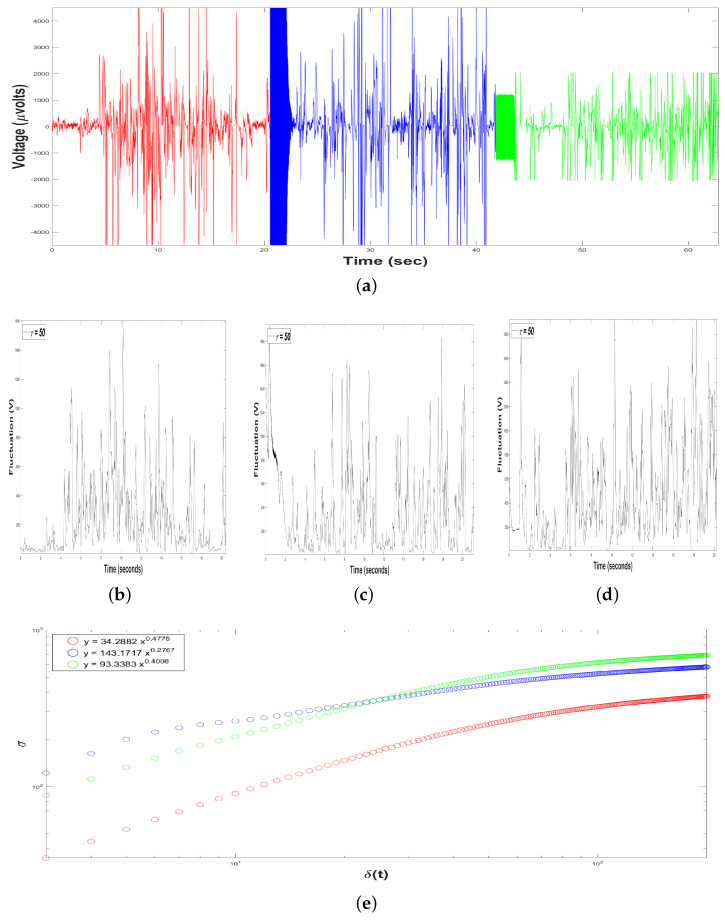
Results of Dolphin EEG brain activity from Experiment Design. (**a**) Representation of the RAW data x(t) (μV vs. sec) for the first measure (in red color), the second measure (in blue color) and the third measure (the green color). (**b**–**d**) Time series of fluctuation of Dolphin EEG brain activity, F(t+τ) for τ=50: for the first measure (**b**), the second measure (**c**) and the third measure (**d**). (**e**) Structure function of fluctuations of Dolphin EEG brain activity, ($\sigma \left(\tau ,\delta_{t}\right)$ vs. $\delta_{t}$) for 3≤τ≤200, for the first measure (in red color), the second measure (in blue color) and the third measure (in green color).

**Figure 6 animals-11-00417-f006:**
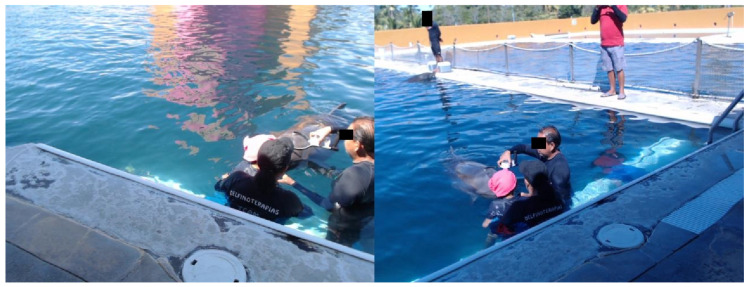
Placement of the device on the dolphin brain to capture the EEG Activity during a Dolphin Assisted Therapy.

**Figure 7 animals-11-00417-f007:**
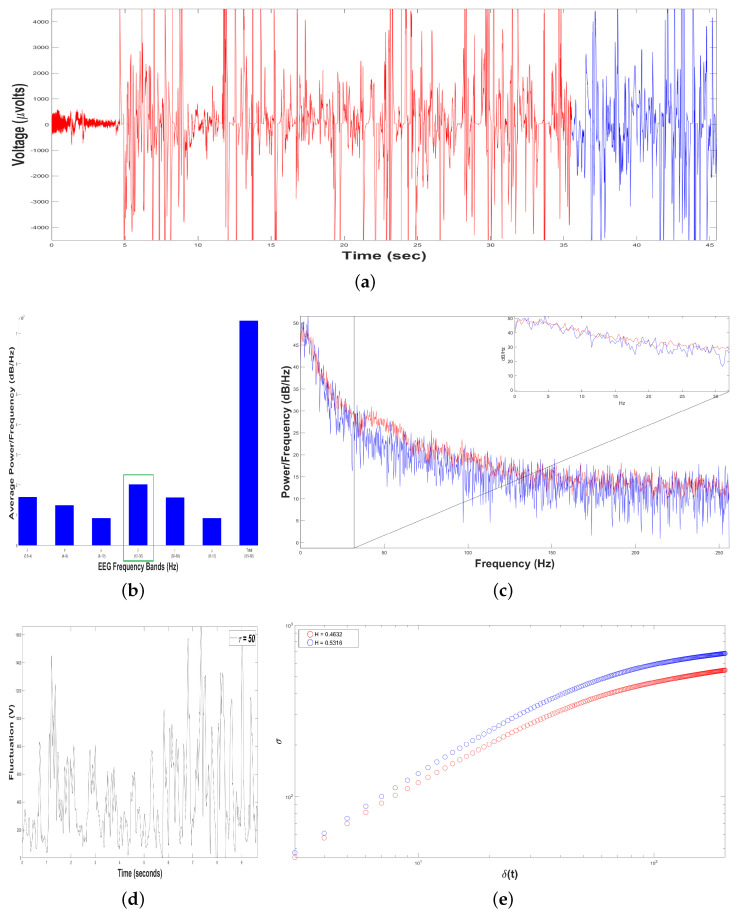
Results of Dolphin EEG brain activity for Intervention Patient 1. (**a**) Representation of the RAW data x(t) (μV vs. sec) for Sample 1 (in red color) and Sample 2 (in blue color). (**b**) Average Power Spectrum Density. (**c**) Power Spectrum Density from 0.5 to 256 Hz, for Sample 1 (in red color) and Sample 2 (in blue color). (**d**) Time series of fluctuation of Dolphin EEG brain activity, F(t+τ) for τ=50. (**e**) Structure function of fluctuations of Dolphin EEG brain activity, ($\sigma \left(\tau ,\delta_{t}\right)$ vs. $\delta_{t}$) for 3≤τ≤200, for Sample 1 (in red color) and Sample 2 (in blue color).

**Figure 8 animals-11-00417-f008:**
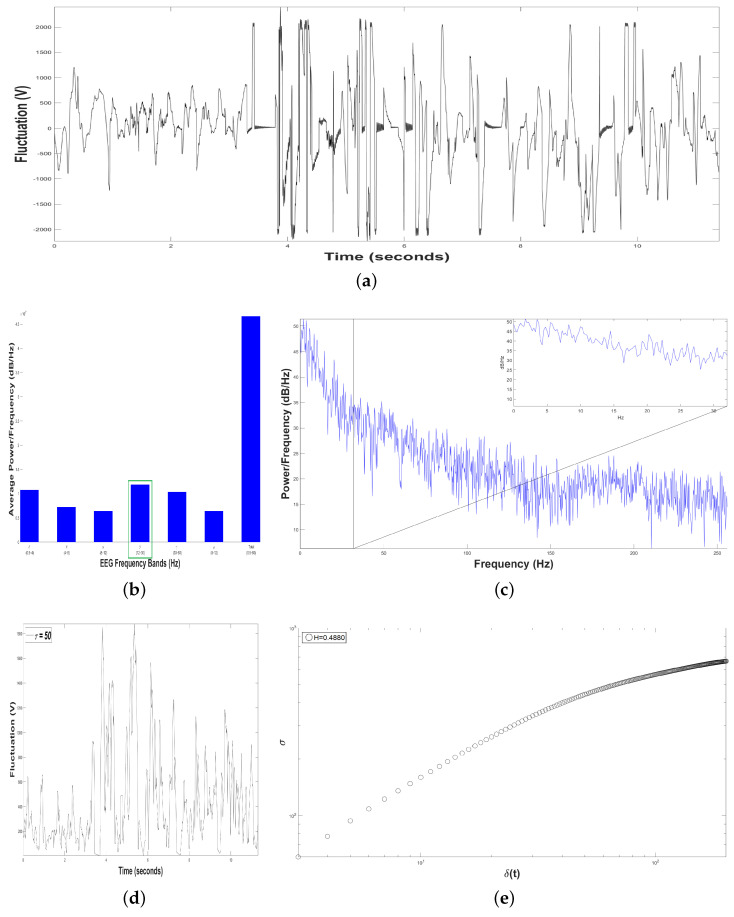
Results of Dolphin EEG brain activity for Intervention Patient 2. (**a**) Representation of the RAW data x(t) (μV vs. sec) During DAT. (**b**) Average Power Spectrum Density. (**c**) Power Spectrum Density from 0.5 to 256 Hz, During DAT. (**d**) Time series of fluctuation of Dolphin EEG brain activity, F(t+τ) for τ=50. (**e**) Structure function of fluctuations of Dolphin EEG brain activity, ($\sigma \left(\tau ,\delta_{t}\right)$ vs. $\delta_{t}$) for 3≤τ≤200, During DAT).

**Figure 9 animals-11-00417-f009:**
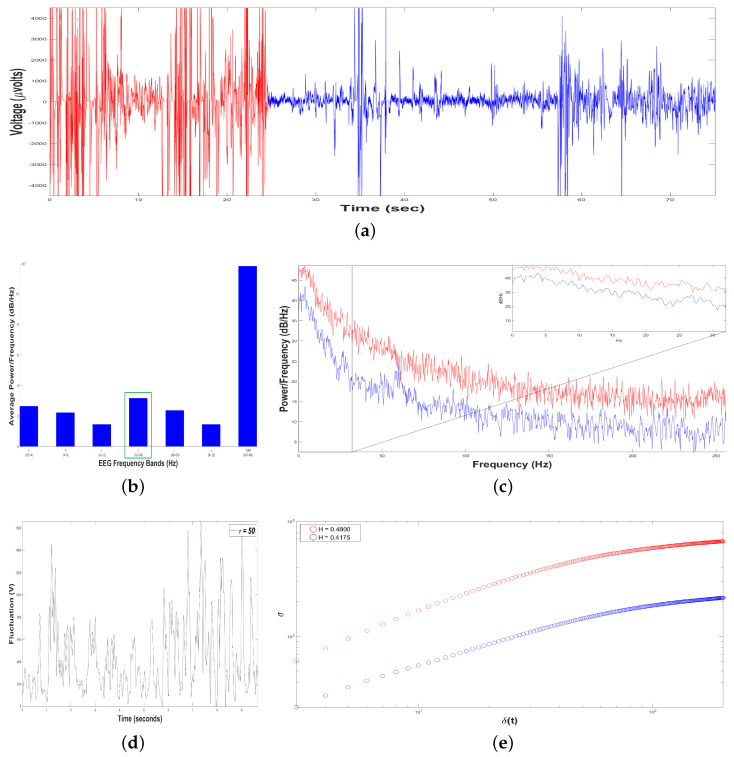
Results of Dolphin EEG brain activity for Intervention Patient 3. (**a**) Representation of the RAW data x(t) (μV vs. sec) for Sample 1 (in red color) and Sample 2 (in blue color). (**b**) Average Power Spectrum Density. (**c**) Power Spectrum Density from 0.5 to 256 Hz, for Sample 1 (in red color) and Sample 2 (in blue color). (**d**) Time series of fluctuation of Dolphin EEG brain activity, F(t+τ) for τ=50. (**e**) Structure function of fluctuations of Dolphin EEG brain activity, ($\sigma \left(\tau ,\delta_{t}\right)$ vs. $\delta_{t}$) for 3≤τ≤200, for Sample 1 (in red color) and Sample 2 (in blue color).

**Figure 10 animals-11-00417-f010:**
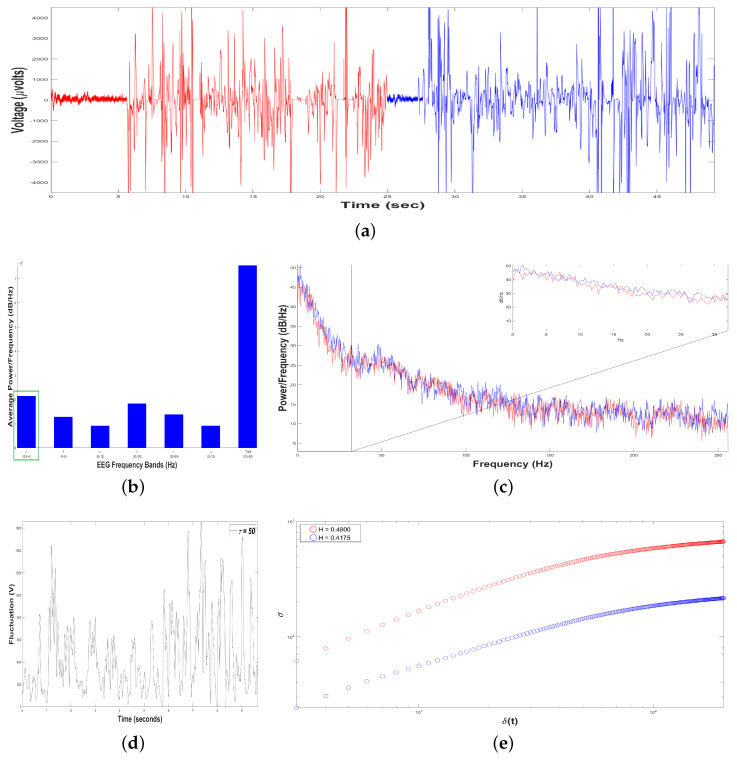
Results of Dolphin EEG brain activity for Control Patient. (**a**) Representation of the RAW data x(t) (μV vs. sec) for Sample 1 (in red color) and Sample 2 (in blue color). (**b**) Average Power Spectrum Density. (**c**) Power Spectrum Density from 0.5 to 256 Hz, for Sample 1 (in red color) and Sample 2 (in blue color). (**d**) Time series of fluctuation of Dolphin EEG brain activity, F(t+τ) for τ=50. (**e**) Structure function of fluctuations of Dolphin EEG brain activity, ($\sigma \left(\tau ,\delta_{t}\right)$ vs. $\delta_{t}$) for 3≤τ≤200, for Sample 1 (in red color) and Sample 2 (in blue color).

**Table 1 animals-11-00417-t001:** Summary of the experimental results of Experimental Design. Low Frequencies 0.5–12 Hz band and High Frequencies 12–60 Hz band. Average values of the scaling exponent *H*.

Sample	Power Spectrum Density dBHz		Self-Affine Analysis	
	Lower Band	Higher Band	*H*	CrossOver
1	$3.3\times 10^{7}$	$2.7\times 10^{7}$	0.4775	192
2	$3.8\times 10^{7}$	$3.7\times 10^{7}$	0.2767	117
3	$5.1\times 10^{7}$	$3.9\times 10^{7}$	0.4008	104

**Table 2 animals-11-00417-t002:** Summary of the experimental results from Experimentation.

During DAT	Power Spectrum Density dBHz		Self-Affine Analysis	
	0.5–4 Hz Band	12–30 Hz Band	*H*	Crossover
Intervention 3	$2.5\times 10^{7}$	$3.1\times 10^{7}$	0.4595	91
Intervention 1	$0.65\times 10^{7}$	$0.75\times 10^{7}$	0.5356	114
Intervention 2	$1.1\times 10^{7}$	$1.3\times 10^{7}$	0.4880	144
Intervention 3	$1.8\times 10^{7}$	$2.3\times 10^{7}$	0.4537	152
Intervention 1	$1.6\times 10^{7}$	$2.0\times 10^{7}$	0.5019	157
Control	$1.75\times 10^{7}$	$1.55\times 10^{7}$	0.4890	183
Control	$2.2\times 10^{7}$	$1.8\times 10^{7}$	0.5251	276

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
