# Peer review of "Non-Parametric Evaluation Methods of the Brain Activity of a Bottlenose Dolphin during an Assisted Therapy"

_animals, 2021, doi:10.3390/ani11020417_

Round 1

Reviewer 1 Report

  1. Please check minor grammar, formatting and spelling errors throughout.
  2. Line 8, simple summary: 'modifies the usual 8 behavior of a female bottle-nose dolphin.' - what are these? Needs 1-2 sentences to elaborate what these changes are and what the implications of these changes are. Please also add this information to the abstract. This should include information on the number of experiments and controls.
  3. Line 15: check correct use of 'on one hand'/ 'on the other hand'
  4. Throughout: spelling Participant in capitals
  5. Line 37: increased hormone release: please be more specific: which hormones/ what is the evidence
  6. From line 36: All of these aspects need references to provide evidence.
  7. References, in general, are sparse: more (and recent) references are required to stress points and provide evidence for statements made.
  8. The research gap needs to be more clearly addressed (i.e. highlighting the need for understanding brain activity in dolphins and what are the possible implications of doing so) - this will require some re-writing of the introduction. Much of the information is also repetitive currently, please make sure this is not the case when re-writing the introduction, ensure its kept concise. 
  9. The introduction currently does not read very well. Please, instead of various subheadings discuss prior relevant research on the focal species (this should include information on bottlenose dolphins, rather than having a separate paragraph), dolphin assisted therapy and then clearly highlight research gap and implications and then information of how this study can address this (this may include hypotheses that relate to research gap)
  10. The bottlenose dolphin section is redundant (see comment above), please only present information that is relevant to the study.
  11. DAT section: see comment number 9. Lacks references.
  12. Methods: please make sure this section is concise and methods/materials are only presented relating directly to your study. Please refer to similar studies to make this more concise.
  13. Please immediately provide an insight in the type and experimental set up including controls. This information is currently getting lost in the sheer volume of additional information provided.
  14. EEG capture device figure is redundant. Figures should show results (not e.g. testing prototype), graphs are currently not up to scientific publication standard, please re-do them with scientific formatting in mind - refer to similar publications. Please remove any figures or images that do not show results (e.g. figures showing time series that do not belong in results - like figures 7, I also don't see any value in figures 1-8)
  15. This paper is not focussing on the method of recording brain waves in bottlenose dolphins - the description of all the tests and prototypes is therefore redundant. Please refer to other publications or cut down to make it more concise. Consider putting certain protocols or steps in appendices.
  16. Remove bullet points from the method, provide information in text.
  17. Results and discussion: needs an introductory/ overview sentence first
  18. Remove bullet points/ add information in text
  19. Figures need to be described in the text to allow the reader to gauge results without having to refer to figures.
  20. Figure legends (e.g. Figure 9) are too small to read, please change for all figures
  21. Why were the specific results (from this specific time series selected)>
  22. Please present and discuss the most important results first, and being really clear about key results and conclusions. 
  23. Please make sure the whole manuscript is concise. e.g. line 672 contains filler sentences that are redundant and muddle the messages.
  24. Line 793: 'sea mammals have advanced communication and a greater power' - please provide evidence for this statement. It is also not clear what exactly is meant by this. Please be really clear about key results and how these related to the literature (even similar work on similar species)
  25. The discussion is very shallow, please provide deeper insights into the key aspect of findings: how to the experiments compare, how does this contrast with the controls? What are possible mechanisms underlying these patterns and what are the implications for this?
  26. How do your results relative to the literature?

Reviewer 2 Report

This article aimed to design a method to assess the potential changes in brain activity in a bottlenose dolphin during Dolphin Assisted Therapies. The article is interesting and there is a high amount of work. However, there is a lack of replication of the experiments and more variables could have been taken into account. Moreover, the manuscript should be proofread before resubmitting it.

Comments

The manuscript is quite long. I would suggest to review the introduction and delete unnecessary paragraphs (e.g. biology of the bottlenose dolphin). There’s also information repeated throughout the manuscript. The text should therefore be reviewed before resubmitting the manuscript.

In the “General discussion”, authors compare the “intervention average” and the “control average”, but a statistical analysis should be done in order to know if there is indeed a difference.

Questions

Were the control experiments and experiments during DAT done in the same environmental conditions (date, time, environment in the pool…)?

Why not including more variables in the analysis (e.g. time, different conditions in the pool…)?

Were the experiments during DAT replicated on several occasions?

Could the author describe the control conditions? Were there before/after measurements?

Round 2

Reviewer 1 Report

Thank you for taking the comments into consideration and making significant improvements to the manuscript, which has increased the readability and its value. Some final, but rather minor comments:

  1. I would suggest having a native English speaker read and edit the manuscript checking for grammar, flow and helping with colloquialisms, as a few issues remain with respect to sentence structure. E.g. line 28-31 could be slightly modified to be more readable, line 107: very long, bulky sentence, line 112 'So....', use of On the one hand (as noted before) - e.g. line 159, use of 'besides' e.g. line 171, line 708 and so on
  2. Referencing needs to be improved - rather than referencing at the end of each paragraph, please consider references after each statement allowing the reader to locate the relevant literature (e.g. in line 32, references of the relevant studies should be behind statements, rather than together at the end of the paragraph.
  3. Line 97: 'did not compromise their integrity in any way' - what does this refer to, please clarify or rewrite.
  4. Figures: deletion of the unnecessary figures have improved the manuscript, but please ensure that labels are readable and clear (and not too small or distorted - e.g. figure 3)
  5. Figure 5: is it possible to provide a clearer figure?
  6. Requires a little restructuring: please do not show/repeat results in the discussion but rather discuss key results and implications. Please make sure that a non-specialist could read this section and understand its implications before having read the whole paper.
